# Enabling work participation for people with musculoskeletal conditions: lessons from work changes imposed by COVID-19: a mixed-method study

LaKrista Morton,[1,2,3] Kevin Stelfox,[1,2,3] Marcus Beasley,[1,2,3] Gareth T Jones [1,2,3], Gary J Macfarlane,[1,2,3] Karen Walker-Bone [4], Rosemary J Hollick [1,2,3]

For numbered affiliations see end of article.

**Correspondence to**
Dr Rosemary J Hollick; rhollick@abdn.ac.uk

## ABSTRACT

**Objectives** To understand what we can learn from the impact of the COVID-19 pandemic and lockdown about what enables work participation for people with inflammatory arthritis and chronic pain conditions.
**Design** Qualitative interviews embedded within an observational questionnaire study of individuals with musculoskeletal (MSK) conditions.
**Setting** UK primary care (general practices), and secondary care-based rheumatology services.
**Participants** Individuals with axial spondyloarthritis, psoriatic arthritis and MSK pain from three established cohorts completed an online/paper-based questionnaire (July–December 2020). A subset of respondents were selected for semistructured interviews.
**Primary and secondary outcome measures** The survey quantified the effects of lockdown on work circumstances. Qualitative interviews explored the impacts of these changes and the advantages and disadvantages of changes in work circumstances.
**Results** 491 people (52% female, median age 49 years) who were employed at the time of lockdown responded to the questionnaire. The qualitative analysis included 157 free-text comments on work from the questionnaire and data collected within 18 interviews.
Participants reported impacts on mental and physical health, and significant financial anxieties. The impact of work changes varied depending on individual and home circumstances. Some felt forced to ignore advice to shield and continue working. The flexibility offered by home working and changes in commuting enabled greater physical activity for some, while others missed the exercise normally undertaken as part of their commute. Others reported a constant need to be 'present' online, which heightened anxiety and worsened MSK symptoms.
**Conclusion** Lockdown showed that flexible working arrangements, which consider the positive and negative aspects of commuting, posture, movement, and work environment matter for work participation, and can have wider benefits in terms of health and well-being for those with long-term MSK conditions. Incorporating these into new models of work will help make the workplace more equitable and inclusive for people with long-term MSK conditions.

### Strengths and limitations of this study

► The study included people with both inflammatory and non-inflammatory musculoskeletal conditions in well-characterised clinical cohorts of 'real world' patients predefined by symptoms or diagnosis.
► Inflammatory arthritis and chronic pain are good exemplars of disability for work caused by a range of long-term conditions and findings may be applicable to a wider range of people trying to work with chronic health conditions.
► A number of respondents worked in professional/ associate professional roles, and fewer of our participants worked in lower paid jobs.
► We collected data from participants at a single time point, and therefore, could not capture longitudinal data, including the longer-term impact of changes to working practices.

## INTRODUCTION

Being in work that is safe, healthy and which gives individuals some control of their work is good for physical, social and psychological health.[1] Employment can contribute to an individual's personal identity, facilitates social relationships, and can bring meaning and purpose in life.[1 2] It also brings better financial control and benefits for dependents.[2] People with chronic musculoskeletal (MSK) conditions (inflammatory rheumatic diseases and chronic pain conditions) want to participate in work[3] but many struggle to be able to do so.[4–7] The pain, fatigue, mobility impairment and functional loss associated with chronic MSK conditions are themselves challenges to work. However, people with MSK conditions also report that factors associated with the nature and type of their employment are also important: line manager and coworker support; flexibility of working hours and practices; time off to attend healthcare appointments; support

for travel to/from the workplace; availability of car-parking; autonomy over how and when to complete work tasks and availability of simple, practical workplace modifications.[3]

In 2020, the global COVID-19 pandemic created unprecedented change to peoples' lives. In the UK, the first case was reported in January 2020, and such was the pace of transmission that a national lockdown was declared from 23 March 2020. People were only permitted to leave home to attend work if they were designated 'key workers' (eg, health and social care, transport, communications, supermarkets). Working changed dramatically: most key workers found their jobs increasingly pressured; those who could work from home made that transition but many other workers were furloughed (paid 80% salary subsidised by government), had to change job or were made redundant.

Therefore, the working lives of most UK workers changed overnight. However, working lives were not the only things that changed during lockdown: elective healthcare was temporarily suspended and then resumed only remotely; access to primary care was hampered; and some people with MSK conditions (particularly those with inflammatory rheumatic conditions treated with immunosuppressive therapies) were advised to 'shield' (stay at home and not go to work, even if they were key workers). Leaving home to exercise was encouraged but restricted to 1 hour/day and most facilities (parks, swimming pools, gyms) were closed. Taking exercise became restricted to walking, running or cycling, all challenging for people with MSK conditions.

The impact of the pandemic does however provide opportunities to evaluate the impact of changed working practices on people with chronic MSK conditions who were working at the time of lockdown, with the objective of learning lessons (good and bad) about supporting work participation for people with these conditions. Within three well-defined cohorts of individuals with inflammatory and non-inflammatory conditions who were in paid employment immediately before the first UK lockdown, we aimed to, first, quantify effects of lockdown on their work circumstances (those who remained in work as usual; those who continued working but from home/elsewhere; and those who were furloughed, changed job or made redundant) and describe the socio-demographic and clinical characteristics of people within each group. Second, we aimed to explore individual's experiences of changes to work circumstances and the advantages and disadvantages of these changes.

## METHODS
### Study design
Qualitative study embedded within an observational questionnaire study of individuals with MSK conditions, which has been reported in line with the Standards for Reporting Qualitative Research (Supplementary File).

### Setting
UK primary care (general practices), and secondary care-based rheumatology services.

### Participants
Participants from three UK cohorts were included in this study (the CONTAIN Study - COvid-19 aNd musculoskeleTal heAlth durIng lockdown). The methods of data collection for this study are published.[8] Briefly, the cohorts comprising the CONTAIN Study included individuals meeting ASAS criteria for axial spondyloarthritis (Assessment of Spondyloarthritis International Society; axSpA[9];) who were originally recruited from 83 clinical rheumatology services (sites) across the UK during 2012–2017 (BSRBR-AS register); patients meeting classification criteria for psoriatic arthritis currently being recruited from sites across the UK (CASPAR[10] - Classification of Psoriatic Arthritis; BSR-PsA register); and people recruited from general practices in three Scottish health boards who had consulted with regional pain in primary care and other symptoms (sleep problems and somatic symptoms) 2016–2017 (MAmMOTH Study[11] - Maintaining Musculoskletal Health Study).

Participants from these three cohorts completed a questionnaire as an additional follow-up for the CONTAIN Study.

### Data collection
#### Questionnaire
Data were collected by questionnaire (online or by paper if requested) starting June 2020, with reminders posted from September 2020. The questionnaire comprised existing validated instruments and questions specific to individuals' experiences during the COVID-19 pandemic.[8] Relevant aspects for the current analysis included socio-demographic characteristics including gender, age and current (main) job and industry. Deprivation status was determined using postcodes with reference to either the population of Scotland,[12] England[13] or Wales.[14] Main job and industry were coded to 4-digit Office for National Statistics Occupational Classification (SOC) Hierarchy codes[15] (table 1). Respondents indicating that they were in employment prior to lockdown were asked whether and how their job had changed since lockdown. They were asked to evaluate their financial security (how difficult they thought it would be to meet their financial commitments 'this month' and 'over the next 12 months' (0–10 Likert scale, 0='not at all worried' and 10='extremely worried'). Written consent was obtained at the time of survey completion.

An open-text question asked about individuals' perceptions and experiences (positive and negative) during the COVID-19 pandemic:

> The situation brought about the COVID-19 pandemic has brought challenges to many of us, but also perhaps some positive changes to day-to-day life. If you wish, please use this box to describe the main

**Table 1** Office for National Statistics Standard Occupational Classification (SOC) examples

| SOC code for quotes | Major SOC category | Examples of occupations and 4-digit SOC codes |
|---|---|---|
| 1 | Managers, directors and senior officials | Chief executive (1115), health service manager (1181), head of public relations (1134) |
| 2 | Professional occupations | Social scientist (2114), civil engineer (2121), information technology professional (2139), primary school teacher (2315) |
| 3 | Associate professional and technical occupations | Laboratory technician (3111), paramedic (3213), artist (3411), careers advisor (3564) |
| 4 | Administrative and secretarial occupations | Post office clerk (4123), receptionist (4216), office manager (4161) |
| 5 | Skilled trades occupations | Landscape gardener (5113), electrician (5241), chef (5434), florist (5443) |
| 6 | Caring, leisure and other service occupations | Teaching assistant (6125), veterinary nurse (6131), hairdresser (6221) |
| 7 | Sales and customer service occupations | Retail assistant (7111), telephone salesperson (7113), window dressers (7125) |
| 8 | Process, plant and machine operatives | Textile process operative (8113), quarry worker (8123), scaffolder (8141) |
| 9 | Elementary occupations | Farm worker (9111), packer/bottler (9134), cleaner (9233) |

challenges that you have faced, and/or any positive changes that you've experienced.

## Qualitative data

We conducted semistructured qualitative interviews with a subset of those who had completed the questionnaire and consented to further contact. Participants were purposively selected for interview across the UK based on information provided in the questionnaire: gender, employment status ((1) those who remained in work as usual, (2) those who continued working but from home elsewhere and (3) those who were furloughed, changed job or made redundant); age, and nature of their MSK condition (inflammatory (axSpA, PsA) and non-inflammatory).

Interviews were carried out by telephone and focused on impacts of the pandemic and restrictions on employment, access to healthcare and on health. For the purposes of the current analysis, we analysed interview transcripts from those who were currently working. Work-related questions specifically aimed to gather insights about how people's work and ability to work had changed because of the pandemic. We also asked about changes to working practices, financial impacts, their employer's awareness of their condition and any concerns about returning to work.

Interviews were conducted by KS, a research fellow with significant experience in conducting qualitative interviews with people with experience of MSK conditions and chronic pain. At the time of scheduling the interview with each potential participant, KS discussed the reasons for doing the research and answered any questions. Consent was obtained prior to interview using a written consent guide and this was audio recorded. Interviews were audio recorded and transcribed verbatim.

In addition, any responses from the open-ended question in the questionnaire that referred to work/ employment were analysed thematically as part of the qualitative analysis.

The qualitative data collected as part of the CONTAIN study provided the opportunity 'to generate a deep understanding of people's experiences, motivations, beliefs, goals, expectations and needs'.[16]

## Data analysis

Of interest to the analysis presented in this manuscript are the CONTAIN study participants who were in paid employment immediately before the first UK lockdown. Using the questionnaire data, we investigated the sociodemographic characteristics of respondents based on their occupational status at the time: (1) those who remained working as usual (including key workers), (2) those who continued working but from home/elsewhere and (3) those who were furloughed, changed job or made redundant. The sociodemographic characteristics of these groups were explored using simple frequencies. Measures of financial concern were investigated across occupational groups.

Transcribed interviews and free-text responses from the questionnaire were uploaded into NVivo V.12 software to facilitate organisation and analysis of the qualitative data. Qualitative data was analysed thematically by KS and LM, supported by RH. Deductive and inductively derived coding[17] was used to identify and categorise themes within higher-order themes informed by the topic guide which specifically aimed to facilitate an understanding of individuals' lived experiences of working with a MSK condition during the pandemic and the benefits and challenges posed to working routines and environments during lockdown. The analytical process involved familiarisation with data and initial coding; organisation of codes according to similarity of meaning; and development and review of themes and subthemes. Emerging analysis was discussed and developed with all authors. Data saturation was deemed to have been achieved through thematic and

code saturation,[18] which was discussed and determined between KS and LM.

Questionnaire data and semistructured interview data were collected and analysed concurrently, to provide an in-depth understanding of lived experiences of people with chronic MSK conditions working during lockdown and to learn, from changes made during the pandemic, about how to best support future work participation.

### Patient and public involvement

Support at work has been identified as a key priority by patients. Patients and patient organisations provided input into items asked in the questionnaire, design of the interview schedule and review of study documentation. Our patient partners also provided comment on the manuscript. We will continue to work with our patient partners to create a summary of findings and disseminate these via our patient organisation partners.

### RESULTS

In total, 1054 individuals completed a CONTAIN study questionnaire (596 from BSRBR-AS, 162 from BSR-PsA, and 296 from MAmMOTH), representing 29% of those contacted (27% BSRBR-AS; 26% BSR-PsA; 33% MAmMOTH). Of the 491 who were in paid employment before lockdown and had complete data, 51.7% were female (0.2% non-binary) with a median age of 49 years (range 21–75); 61.9% were from BSRBR-AS, 17.9% from BSRBR-PsA and 20.2% from MAmMOTH.

We included 157 responses from the free-text questionnaire item in the qualitative thematic analysis that referred to 'work'. 57.3% of these respondents were from BSRBR-AS, 19.1% from BSR-PsA and 23.6% from MAmMOTH. 57.3% were female and had a median age of 47 years (range 27–78). Of 782 questionnaire respondents who provided consent to be contacted about an interview, we interviewed 23 (18 of whom were in paid work immediately prior to the start of the first UK lockdown and therefore included in this qualitative analysis) and reached data saturation at this stage. Of the 18 interviewees who were in paid work immediately prior to the start of the first UK lockdown: 7 were from BSRBR-AS, 7 from BSR-PsA and 4 from MAmMOTH. Eight were female, they were aged 28–64 years and lived in England (n=12), Scotland (n=5) and Wales (n=1). Of these 16 were Managers, directors (senior officials), professionals or in associate professional and technical occupations (table 1).

Key themes identified within qualitative interviews and free-text questionnaire item responses are described within table 2.

### Changes to work status and circumstances

Table 3 summarises the changes to work status caused by lockdown of working questionnaire respondents. In total, 55% (n=268) continued to work as usual (most of whom, 76% (n=205) were key workers), 24% (n=120) changed to home working, and 21% (n=103) changed job/were furloughed/ made redundant (table 3). Older workers (aged >56 years)

**Table 2** Key themes identified within qualitative interviews and free-text questionnaire item responses

| Primary areas of inquiry | Themes |
|---|---|
| Changes to work status and circumstances | ► Decisions about remaining at work<br>► Decisions about treatments<br>► Loss of earning<br>► Flexibility |
| Disadvantages and advantages of changes to work circumstances | ► Managing multiple roles within the family<br>► Maintaining physical activity<br>► Working at home<br>► Workstation set up and ergonomics<br>► Relationship with employer<br>► Making adaptations<br>► Changes in pace of life<br>► Social interactions<br>► Stigma |

were those most likely to have been furloughed/changed job or made redundant (43.7%) and unsurprisingly no one working as process/plant/machine operatives, skilled tradesperson or in an elementary occupation was able to work from home.

The interviews provided insight into the changes to work status and the impact of these changes to health and financial circumstances. Some felt they had no choice but to continue working as usual, despite advice to shield.

> Me and my family being forced to work whilst most people were at home was stressful, my work not implementing safety procedures early or well enough was also a challenge. The only positive is that we haven't caught it. Q499 (Male; 40–55; SOC1)

A community care worker described how her company were unable to furlough her, despite advice to shield, which left her with no choice but to return to work.

> Quite a few of the people I go to, like I say, they're elderly and they still have family members coming in and out to do their shopping. So, the people I've been going to haven't been fully isolating they haven't been able to so, just been fortunate I haven't picked it up. I16 (Female;≤39; SOC6)

Others stopped taking their immunosuppressant medication (prescribed to control their inflammatory arthritis) so that they could continue working.

> I stopped my medication for 2 months as I was very wary about taking it, I spoke to my rheumatologist first, I was also then allowed to continue to work, my employers were very good and we had gone down to just 3 members of staff. Questionnaire 360 (Female; 40–55; SOC3)

For some, being self-employed enabled them to continue working in a way that suited them and their MSK condition.

I think because obviously I run my business, I've been able to be really flexible and work to suit myself, where if I'd been employed I don't know how that would have impacted me. I11 (Female; 40–55; SOC1)

Those who had lost their jobs explained that choices about alternative work would be influenced by their MSK condition, and might make them vulnerable to financial hardship.

Unfortunately I was made redundant due to COVID-19 and my work closed. I am looking to become self-employed but worried about money and if I can work as hard as I can. Q375 (Male; 40–55; Formerly SOC 4)

When asked about financial stability within the questionnaire, we found a wide distribution of responses, with generally more concerns expressed about finances in 12 months' time as compared with next month (table 4).

Qualitative data collected from questionnaire free-text responses and the semi-structured interviews indicated that those who were unable to work from home generally reported more economic anxiety.

they [employer] wouldn't furlough me, so they gave me… it was part sick pay, type thing…because I'm part time I don't earn enough to get full sick pay. I16 (Female;≤39; SOC6)

Some respondents reported profound financial impact, exacerbating existing health worries and creating additional health problems.

The main challenge for me has been trying to survive financially. My job ended as soon as the lockdown was announced and I received only one final wage. I have not been eating properly as I cannot afford to and this is making me very depressed. Q478 (Female,≤39, SOC4).

I've had hospital appointments cancelled, new medication cancelled, financial loss of up to a third per month (approx £700 down per month) depression, severe weight gain. Q19 (Male;≤39; SOC5)

### Disadvantages and advantages of changes to work circumstances

Home working offered the opportunity for flexibility but the ability to take advantage of this depended on

**Table 3** Changes to individuals' work due to the pandemic, by sociodemographic factors (n=491)

| | | Continued working as usual, n (%); *Keyworkers, n (%)* | | Working from home, n (%) | Furloughed/made redundant/ changed job, n (%) |
|---|---|---|---|---|---|
| Gender | Male | 126 (47.0) | *87 (42.4)* | 54 (45.0) | 56 (54.4) |
| | Female | 141 (52.6) | *117 (57.1)* | 66 (55.0) | 47 (45.6) |
| | Non-binary | 1 (0.4) | *1 (0.5)* | 0 | 0 |
| Age | 39 and under | 56 (20.9) | *45 (22.0)* | 33 (27.5) | 23 (22.3) |
| | 40 to 55 | 133 (49.6) | *104 (50.7)* | 60 (50.0) | 35 (34.0) |
| | 56 and over | 79 (29.5) | *56 (27.3)* | 27 (22.5) | 45 (43.7) |
| Job type | Managers, directors, and senior officials | 31 (11.6) | *16 (7.8)* | 20 (16.7) | 18 (17.5) |
| | Professional occupations | 98 (36.6) | *85 (41.5)* | 55 (45.8) | 16 (15.5) |
| | Associate professional and technical | 32 (11.9) | *23 (11.2)* | 17 (14.2) | 17 (16.5) |
| | Administrative and secretarial | 23 (8.6) | *18 (8.8)* | 25 (20.8) | 7 (6.8) |
| | Skilled trades occupations | 19 (7.1) | *9 (4.4)* | 0 | 14 (13.6) |
| | Caring, leisure and other services | 23 (8.6) | *20 (9.8)* | 1 (0.8) | 7 (6.8) |
| | Sales and customer service | 10 (3.7) | *9 (4.4)* | 2 (1.6) | 8 (7.8) |
| | Process, plant and machine operatives | 13 (4.9) | *10 (4.9)* | 0 | 9 (8.7) |
| | Elementary Occupations | 18 (6.7) | *15 (7.3)* | 0 | 7 (6.8) |
| | Missing/NA | 1 (0.4) | *0* | 0 | 0 |
| Deprivation | 1—Most deprived | 28 (10.4) | *24 (11.7)* | 4 (3.3) | 11 (10.7) |
| | 2 | 37 (13.8) | *27 (13.2)* | 14 (11.7) | 15 (14.6) |
| | 3 | 57 (21.3) | *44 (21.5)* | 25 (20.8) | 28 (27.2) |
| | 4 | 80 (29.9) | *61 (29.8)* | 32 (26.7) | 23 (22.3) |
| | 5—Least deprived | 66 (24.6) | *49 (23.9)* | 45 (37.5) | 26 (25.2) |
| Total n | | 268 | *205* | 120 | 103 |

NA, not available.

**Table 4** Perceived difficulty and worry about meeting financial commitments in the coming month and over the next 12 months by job type (n=520)

| Occupation (n) | Median score (IQR) | |
|---|---|---|
| | Perceived difficulty to meet financial commitments 'this month' (0=not at all difficult; 10=extremely difficult) | Worry about meeting financial commitments over the next 12 months (0=not at all worried; 10=extremely worried) |
| Managers, directors and senior officials (72) | 0 (0–1) | 2 (0–4) |
| Professional occupations (173) | 0 (0–1) | 1 (0–3) |
| Associate professionals and technical occupations (71) | 1 (1–2) | 2 (0–6) |
| Administrative and secretarial occupations (56) | 1 (0–1) | 1 (0–3) |
| Skilled trades occupations (36) | 0.5 (0–4.5) | 2 (0–5.5) |
| Caring, leisure and other service occupations (38) | 0 (0–3) | 2 (0–5) |
| Sales and customer service occupations (24) | 1 (0–2) | 2 (0.5–5) |
| Process, plant and machine operatives (23) | 0 (0–6) | 1 (0–6) |
| Elementary occupations (27) | 0 (0–2) | 0 (0–5) |

individual and home circumstances. Those with partners who were working, for example, as key workers, and those with caring responsibilities, described difficulties juggling family and work responsibilities on top of their arthritis and recommendations to self-isolate.

> The main challenge has been being stuck indoors or with limited access to outdoor space for months on end whilst at the same time looking after children (my wife is a key worker and has been at work), working a full-time job from home and dealing with my arthritis. I am certain that my arthritis is much more painful now in my hands, arms, shoulders, neck and back because of a poor working from home setup and also the lack of exercise I been able to take. Q355 (Male; 40–55; SOC1)

Others who had previously used their commute to work as an opportunity for exercise were similarly affected by a more sedentary lifestyle, which had a negative impact on their MSK condition.

> I've realised if you cycle to work, you don't feel so stiff at the end of it…if I did nothing, if I came down [stairs], started work, I would just stay stiff all day I think. I10 (Male;≥56; SOC2).

Home workers reflected on work activities being more monotonous with an increase in more repetitive, computer-based tasks. For some, this led to an overall reduction in physical activity and more pain.

> I'm not getting up and down as much as I would in the practice, I would be up and asking the GP a question, going to reception, go out to get a patient, having a chat with a nurse, so all of that movement has disappeared and I'm sitting all day…the lack of exercise probably has impacted more I think. I4 (Female; 40–55; SOC2)

Home working also highlighted the role of trust between employees and employers. For some, not being physically present in the workplace environment created a sense of needing to be constantly 'present online' which negatively impacted on their MSK condition.

> I'm finding myself sitting at my computer earlier and leaving it later at the end of the day…I've noticed that throughout a lot of the members of my team as well, they feel that if an email comes through you need to respond to it quickly or else people think you're being a slacker and that you're not at your desk and you're not working and stuff. I6 (Male;≥56; SOC3)

Several participants also highlighted missing the social aspect of going to work, particularly for those who lived alone and were required to shield.

> Living by myself in isolation has been a real test of my mental strength. Working from home and being without my work colleagues has also been challenging. Lack of social intersection has been the biggest challenge for me. Q9 (Female;≤39; SOC2)

In contrast, some individuals, for example, those without home-schooling commitments, found it easier to take advantage of the flexibility offered by home working. For some, less time spent commuting also provided several benefits such as reducing stress levels, freeing up time for other things including exercise, and improving arthritis symptoms, particularly back symptoms.

> By not driving for 2 hours a day I got more of my work done and was able to walk for an hour a day which has helped my back and wellbeing. Q137 (Female; 40–55; SOC2)

A slower pace of life was helpful for many, facilitating regular rest and energy conservation and improving fatigue while still enabling them to do their job from home.

> I've also been able to have hot water bottles and rest regularly when working at home whilst still being able

to do my job. It slowed my life down in a positive way and made me realise I'd been rushing around trying to fit too much into each day before lockdown. Q289 (Female;≤39; SOC1)

I found shielding for 12 weeks enabled me to have more energy every day. It also enabled me to do some home exercise without the feeling of being too exhausted from being at work. I felt I pushed myself a lot but in a great way. I feel now I am back at work, I am getting back into my old pattern and feel fatigued. Q6 (Female;≤39; SOC 4).

Changing to home working suddenly meant that some did not have suitable equipment, so that they could not work effectively at their computer for prolonged periods, and developed increased pain and fatigue. However, people reported that they gradually developed different ways of working at home which facilitated regular movement throughout the day, and/or scheduled physical activity into their day, which was beneficial, but this required a conscious effort and took time.

I was kind of 'hot desking', whether that was in the kitchen or in the garden or if it was… it's good in some respects, either I was constantly moving around, so therefore it did my back a lot of good. I13 (Male;≤39; SOC2)

One of the real nice things was I was able to go out for lunch with my daughter and we'd go on like a bike ride of something like that. We'd spend an hour out, which I didn't do during a normal working day which is something I think I'll take forward from it, is the stepping out of the building, there's massive sort of benefits for you. I13 (Male;≤39; SOC2)

While some individuals missed social interactions at work, others felt that less contact with their colleagues and/or the work environment reduced their stress levels. In addition, some reflected that home working made their underlying health conditions less visible and hence, less stigmatising.

There are fewer people around=less stressed. I can park my car when I use it for work. I saw less of my irritating colleagues. Q467 (Non-binary; 40–55; SOC2)

I was suffering from mental health issues, and this made me needing to work from home and take time off…[I now] stand out less from my colleagues. Q39 (Male; 40–55; SOC2)

## DISCUSSION

In this study of people living and working with long-term MSK conditions, we have explored the effects of the lockdown caused by COVID-19 on changes in work status and circumstances, and the perceived impact of these changes on health and well-being. People who continued working often reported significant anxiety about becoming infected at work and perceived unfairness compared with those who were furloughed, or could work at home. Many reported anxiety about finances and future chances of employment. In many cases, people found themselves working from home for the first time. Home working was not a panacea for all but offered some advantages and disadvantages in terms of impact on mental and physical health and physical activity. Most importantly, our findings point to some solutions to address the disability employment gap as we move beyond the pandemic to enable those working with MSK conditions to remain in work: suitable equipment; hybrid home working; flexibility; relationship with and support from managers; and minimising the need to travel in traditional commuter times.

There are strengths and weaknesses to consider when interpreting these findings. 61% of questionnaire respondents and 16 interviewees worked in professional/associate professional roles, and fewer of our participants worked in lower paid jobs. We collected data over a 6-month period which reflected varying degrees of COVID-19 public health measures both over that time period and based on where people lived. We did not capture longitudinal data, including the longer-term impact of changes to working practices. However, the study included people with both inflammatory and non-inflammatory MSK conditions in well-characterised cohorts of 'real-world' patients[19] predefined by symptoms or diagnosis as opposed to convenience samples. Inflammatory arthritis and chronic pain are good exemplars of disability for work caused by the range of long-term conditions, and therefore, these findings may be applicable to a wider range of people trying to work with chronic health conditions.

The COVID-19 pandemic has brought into sharp focus population inequalities in terms of the health, work, and finances of people.[20] We found that, in those with MSK conditions, all these impacts coalesced, causing a complex relationship between socioeconomic status, vulnerability to COVID-19 and risk of work-related exposure. As in other studies, individuals aged >56 years made up the highest proportion of people who had changed job/were furloughed/made redundant during the COVID-19 pandemic.[21] This may have important implications for older people with MSK conditions wishing to return to work, as older workers in particular report more difficulties gaining employment after losing a job.[21] Other general population studies have found that those working in more manual/lower paid work were less likely to be able to work from home.[22] However, for individuals with MSK conditions on immunosuppressant medication who were advised to shield and work from home, we have shown those working in manual, lower paid jobs (often public-facing roles with a higher risk of work-based exposure) were less likely to be able to do so. The financial impacts of this were significant if individuals could not be furloughed. People reported having no choice but to continue working and described the anxiety caused in consequence. Recent evidence indicates increased

work-related exposure to COVID-19 in those unable to work from home and who are in closer proximity to other people or in direct contact with the public; often low-income jobs in service sectors, such as health or social care, transportation, cleaning and hospitality.[23] A recent online survey of 2003 disabled workers or workers who have a health condition or impairment and who were in work at the start of the pandemic in February 2021 by the Trades Union Congress,[24] suggests that disability employment gap has widened as a result of COVID-19.

Many of the advantages and disadvantages of working from home highlighted by people with chronic MSK conditions in this study are similar to those reported by workers without health conditions during the pandemic.[25] People learnt to adapt and many were able to work effectively from home. People with MSK conditions valued flexibility to organise tasks and the freedom to make decisions about when they did their work from home. Similarly, 9 out of 10 workers in one national survey reported that they got at least as much, if not more, work done at home as in the office, with almost three quarters of employees saying that they wanted to adopt hybrid working arrangements in the future.[25] For people with MSK conditions, we have already shown that commuting to work, and driving, can be a significant challenge for people with inflammatory arthritis,[7 26] and these findings suggest that, for many, reduced need for commuting was beneficial.

However, home working was not a panacea. Among those for whom exercise formed part of their daily commute, working from home worsened pain and stiffness. Moreover, some missed the sociability and benefits of collaboration offered by working in shared workspaces.[25] While many reported improvements in work–life balance, and more time for exercise, others struggled to balance working from home with domestic responsibilities and managing their MSK condition on top of this created additional burden. Interestingly, general population studies of workers have suggested people developed more MSK pain, and higher levels of fatigue and poor sleep early after lockdown, although things improved somewhat after a period of adjustment.[25 27] Our study also highlighted the importance of the relationship between home-working employees and their managers. In a recent study of UK workers, participants who had more frequent contact with their line manager and had a work station risk assessment while home-working reported better mental health and less MSK pain.[25] Line managers play a key role in supporting, motivating and engaging a remote or hybrid workforce, and employee well-being and line manager support is closely linked to productivity at work.[25]

We already know that if people are out of the workplace for a long period that they are less likely to come back.[28 29] Loss of both routine and contact with the workplace has a detrimental impact on inactivity, isolation, well-being, as well as impaired self-image and career opportunities, physical fitness and confidence.[30] Similarly, prolonged unemployment, for any reason, causes additional health problems. Those who lose their job suffer from worse mental health,[31] poorer life expectancy,[32] attend healthcare consultations more frequently with physical symptoms and report higher levels of pain.[33] The significant numbers of people with long-term MSK conditions who were previously coping at work, and who have now found themselves out of the workplace for long periods because of the pandemic, may suffer similar consequences without additional support.

## CONCLUSIONS

This study provides new insights into the impact of the COVID-19 pandemic on people living and working with MSK conditions within well-defined cohorts. It provides lessons to support those working with long-term MSK conditions, those with other long-term conditions and the wider working population to work well and remain in work. Flexible working arrangements such as home-working can not only make the workplace more inclusive for people with disabilities and health conditions but can also have wider benefits on their MSK health and well-being. These findings should encourage structural and organisational changes at the workplace to support people with long-term MSK conditions to work. There have, however, been concerns that individualised or complicated working patterns might not be sustainable within an organisation or might be perceived as unfair by coworkers.[25 34] However, flexibility and trust between employers and employees are the foundations of 'good work'[35] and employers creating an open, flexible workplace experience better productivity and reduced staff turnover. The lockdown has provided a learning opportunity for both employees and employers to think creatively and shape new models of work that can accommodate everyone. The current study provides evidence of the value of some of these approaches when they are made available.

The pandemic has exposed pre-existing inequalities in socioeconomic circumstances, health and work. Unfortunately, we have seen that people with long-term MSK conditions for whom flexibility was not possible, tended to be those from more socio-economically deprived backgrounds who have consequently been exposed to financial hardship or job loss or felt forced to ignore advice to shield and continue working, putting themselves at increased risk. Rheumatology services may see the consequences of this on work, health and finances for many years to come. However, some of the findings reported here may enable the rheumatology multidisciplinary team to better support work participation for all patients with long-term MSK conditions and address the disability employment gap.

**Author affiliations**
[1]Epidemiology Group, University of Aberdeen School of Medicine Medical Sciences and Nutrition, Aberdeen, UK
[2]Medical Research Council Versus Arthritis Centre for Musculoskeletal Health and Work, University of Aberdeen School of Medicine Medical Sciences and Nutrition, Aberdeen, UK
[3]Aberdeen Centre for Arthritis and Musculoskeletal Health (Epidemiology Group), University of Aberdeen School of Medicine Medical Sciences and Nutrition, Aberdeen, UK
[4]Medical Research Council Versus Arthritis Centre for Musculoskeletal Health and Work, University of Southampton, Southampton, UK

**Contributors** We are grateful to our patient partner Inga Wood for help with designing the interview schedule and for commenting on the manuscript and Lynne Laidlaw for help with designing questionnaire. The authors do not report any conflicts of interest. GJM conceived the idea for the study and all authors were involved in the detailed planning. LM, KS and RJH conducted the qualitative analysis with input from KW-B. MB and GTJ undertook the questionnaire analysis. LM and RJH integrated questionnaire and quantitative findings, and KW-B, KS and GJM contributed to interpretation of findings. RJH and LM drafted the manuscript and all authors contributed important intellectual content via written comments. GJM and RH are responsible for the overall content as guarantors.

**Funding** This work was supported by versus Arthritis (grant number: 20748) and the British Society for Rheumatology. The funding for the original studies included were from versus Arthritis (MAmMOTH) and the British Society for Rheumatology (BSRBR-AS and BSR-PsA). LM is funded through the Medical Research Council/vs Arthritis Centre for Musculoskeletal Health and Work (vs Arthritis grant no. 20665). LM, Research Fellow, PhD; KS, Research Fellow, PhD; MB, Study Co-ordinator, PhD; GTJ, Reader, PhD; GJM, Clinical Chair in Epidemiology, Dean of Interdisciplinary Research and Research Impact, PhD; KW-B, Professor in Occupational Rheumatology, Honorary Consultant in Rheumatology, Director vs Arthritis/MRC Centre for Musculoskeletal Health and Work, PhD; RJH, Senior Clinical Lecturer, Honorary Consultant in Rheumatology, PhD.

**Competing interests** None declared.

**Patient and public involvement** Patients and/or the public were involved in the design, or conduct, or reporting, or dissemination plans of this research. Refer to the Methods section for further details.

**Patient consent for publication** Not applicable.

**Ethics approval** Ethical approval for BSRBR-AS was from NRES Committee North East (County Durham and Tees Valley, Reference 11/NE/0374); BSR-PsA from West of Scotland REC 3 (Reference 18/WS/0126) and; MAmMOTH from NRES Committee South West (Cornwall and Plymouth, Reference 16/SW/0019). Informed consent was given by participants for publication of material.

**Provenance and peer review** Not commissioned; externally peer reviewed.

**Data availability statement** Data may be obtained from a third party and are not publicly available. The data within the article which relate to the collection of BSR register data are owned by the BSR—access to these data are subject to application being made to the BSR: Registers (rheumatology.org.uk).

**ORCID iDs**
Gareth T Jones http://orcid.org/0000-0003-0016-7591
Karen Walker-Bone http://orcid.org/0000-0002-5992-1459
Rosemary J Hollick http://orcid.org/0000-0001-6558-7189

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
