## [Reviewer comments · BMJ Open]

ARTICLE DETAILS

TITLE (PROVISIONAL)	Enabling work participation for people with musculoskeletal conditions. Lessons from work changes imposed by COVID-19: a mixed method study
AUTHORS	Morton, Lakrista; Stelfox, Kevin; Beasley, Marcus; Jones, Gareth; Macfarlane, Gary; Walker-bone, Karen; Hollick, Rosemary

VERSION 1 – REVIEW

REVIEWER	Stijn De Baets Ghent University, Department of rehabilitation sciences
REVIEW RETURNED	04-Nov-2021

GENERAL COMMENTS	See comments in attached PDF documents. General remarks: - Please review scientific language - Describe the methods / results more clear. It looks like the manuscript is now a mix between a qualitative and quantitative report. It is not toally clear what way you want to go. The reviewer provided a marked copy with additional comments. Please contact the publisher for full details.
---

REVIEWER	Fatih Karaarslan University of Health Sciences, Ankara
REVIEW RETURNED	06-Dec-2021

GENERAL COMMENTS	It is not understandable and some abbreviations need to be explain (e.g. MSK in abstract). It could be better to rearrenge whole manuscript. Kind regards
---

VERSION 1 – AUTHOR RESPONSE

Reviewer 1 (Dr. Stijn De Baets, Ghent University):

General Remarks

- 1. Please review scientific language.**

We have reviewed the manuscript and, where appropriate, changes were made to ensure the language is clear. Please also see our response to point 31 regarding use of the first person. Use of “we” is not uncommon in academic writing, when referring to your own body of work (*sf* Sloan et al., 2022; Smith et al., 2021; Hollick et al., 2020). In addition, the passive voice does not represent the way qualitative research is an active process. We have checked the style of other BMJ publications, and our approach appears to be consistent with style used.

2. Describe the methods / results more clear. It looks like the manuscript is now a mix between a qualitative and quantitative report. It is not toally clear what way you want to go.

This is a mixed method study, with qualitative interviews embedded within an observational questionnaire study. The quantitative survey responses provide context for the qualitative data and was used to inform purposive sampling for the interviews. We have added additional subsections within the abstract to clarify the methods. As previously specified within the manuscript, we have added to the abstract that the present study comprises: “*Qualitative interviews embedded within an observational questionnaire study of individuals with musculoskeletal (MSK) conditions.*”

Manuscript Comments

3. General feedback: Please read the text on language and textual mistakes

We have reviewed the manuscript and, where appropriate, made changes to ensure the language is clear.

4. Please provide more details of the methods (abstract)

We have made revisions to the abstract and have provided detail of these changes in our response to the editor (point 1).

5. It would be interesting if some literature regarding the 'meaning' of work can be added

We have added further relevant detail to this section of the introduction.

6. The research question is not a full research questions. Please include all elements of PICOS in de the research question.

As all elements of PICOS are not relevant for the qualitative study described in this manuscript (e.g., intervention) we have rephrased the research question to incorporate those that are relevant.

7. Wondering what kind of design this study is. On the one hand, some quantitative data are provided, on the other, results of open questions and phone interviews. Is this a qualitative research project? Is it a quantitative? At current, it seems like a combination of the 2, where none of them s fully explored. My advice: have a detailed look at the methods of this study, and provide a clear approach. It looks like a convergent mixed method study was conducted? but why don't you do analyse the numeric data a bit more in detail then? add some statistics? At current, the method in insufficient and unclear.

This is a mixed method study, with a qualitative study embedded within an observational questionnaire study. The quantitative data provided the context and sampling frame to explore in depth within the qualitative study the reasons for change in work status, and disadvantages and advantages of changes to work during the pandemic for people with long-term MSK conditions. In the abstract and page 6 of the manuscript we describe the study design as a “Qualitative study embedded within an observational questionnaire study of individuals with musculoskeletal (MSK) conditions.”

Importantly, the descriptive quantitative methods we chose are concordant with the aims of the research; specifically, to quantify changes in work circumstances (those who remained in work as usual, those who continued working but from home /elsewhere, and those who were furloughed, changed job or made redundant) and to describe the socio-demographic and clinical features of each group. This information provides the context of changes to work status during the pandemic, aspects of which are explored fully within the qualitative analysis. We do not think that additional statistics are justified given the aim, nor do they provide additional insights. We have clarified within the study aims the purpose of the quantitative and qualitative analysis in the introduction;

Within three well-defined cohorts of individuals with inflammatory and non-inflammatory conditions who were in paid employment immediately before the first UK lockdown, we aimed to, firstly, quantify effects of lockdown on their work circumstances (those who remained in work as usual; those who

continued working but from home/elsewhere; and those who were furloughed, changed job or made redundant) and describe the socio-demographic and clinical characteristics of people within each group. Secondly, we aimed to explore individual's experiences of changes to work circumstances and the advantages and disadvantages of these changes.

Data on clinical condition and socio-demographic factors collected within the questionnaire were used to purposively select our interview sample to reflect a range of different experiences during the pandemic. We have revised the manuscript to make the criteria for purposive sampling clear (see Methods; Qualitative data). See also response to point 12 below.

8. Please provide more details regarding the setting. Is it a primary physician practice? ...

This information is provided within the "Participants" section within the Methods.

9. Not totally clear: did you recontact these persons? If yes, please describe clearly. Who completed the questionnaire? the people from the original study? please elaborate in clear way. Sampling and selection?

We have added further detail in the Methods to specify that the CONTAIN Study comprised an additional follow-up for the three established cohorts described in the "Participants" section.

10. what were the criteria for purposive sampling? i can only read general criteria that do not allow selection. please elaborate.

Data on clinical condition (i.e., inflammatory (axSpA, PsA) or non-inflammatory MSK condition) and socio-demographic (age, gender, employment status) factors were collected within the questionnaire. This data was used to purposively select our interview sample to reflect a range of different experiences during the pandemic. We have revised the manuscript to make the criteria for purposive sampling clear (see Methods; Qualitative data).

11. what kind of questionnaire? please provide information. was it self-developed? standardized? aiming at ...How did you gather the data? online platform? Paper? ...Period? How did you analyse this?

We have added detail describing that the questionnaire comprised existing validated instruments and questions specific to individuals' experiences during the pandemic, and that data was collected online or by paper if requested. We have added the reference from an already published manuscript from the CONTAIN study to the Methods so that readers can be pointed to further details of the other instruments included in the questionnaire.

12. Was this part of a structured approach? Did you analyse these textual aspects in structured way? how did you approach this? part of the thematic analysis?

Yes, we analysed the free text responses to the open-ended question using a thematic analysis as described in the "Data collection and analysis" sections within the Methods:

Any free-text responses that referred to work/employment were analysed as part of the qualitative analysis.

Within the "Data analysis" section within the Methods we then go on to say:

Transcribed interviews and free-text responses from the questionnaire were uploaded into NVivo 12 software to facilitate organisation and thematic analysis of the qualitative data.

We then proceed to describe the thematic analysis of this qualitative data (see point 18).

13. who conducted the interviews? how did you select the people to be called? Did everyone participate? please add more information

Details about the interviewer and further information about the selection of participants for interview from all those who were eligible has been added to the “Data Collection” section.

As highlighted in response to point 12, we used work status, socio-demographic and clinical characteristics identified from the survey responses to purposively select people to contact for interview.

Within the results, we have added further detail about those we interviewed in relation to the questionnaire sample:

Of 782 questionnaire respondents who provided consent to be contacted about an interview, we interviewed 23 (18 of whom were in paid work immediately prior to the start of the first UK lockdown and were therefore included in this qualitative analysis) and reached data saturation at this stage.

14. should be described clearly in the research question. This is not the case now, please elaborate.

Information about participants’ working status has been added to the research question.

15. Why only descriptives? Please provide a more sincere analysis.

As highlighted in our response to point 9, we have presented a quantitative analysis which is concordant with the aims of the research; specifically, to quantify changes in work circumstances (those who remained in work as usual, those who continued working but from home /elsewhere, and those who were furloughed, changed job, or made redundant) and to describe the socio-demographic and clinical features of each group. This information provides the context of changes to work status during the pandemic, aspects of which are explored fully within the qualitative analysis.

16. Please provide more info on the steps of the thematic analysis, and add literature (e.g. Braun and Clarke?)

The following steps of the thematic analysis, with references from Braun and Clarke, are provided in the Methods section:

Qualitative data was analysed thematically by KS and LM, supported by RH. Deductive and inductively derived coding (17) was used to identify and categorise themes within higher-order themes informed from the topic guide which specifically aimed to facilitate an understanding of 1) individuals’ lived experiences of working with a musculoskeletal condition during the pandemic and 2) to understand, from individuals’ perspectives, the benefits and challenges posed to working routines and environments during lockdown. The analytical process involved familiarisation with data and initial coding; organisation of codes according to similarity of meaning; and development and review of themes and subthemes. Emerging analysis was discussed and developed with all authors. Data saturation was deemed to have been achieved through thematic and code-saturation (18).

17. these 2 goals should not be mentioned here, but should be described in the introduction - research question.

Thank you for this suggestion. We have clarified this within our research aims as requested. However, we would prefer to also leave them here to make the link between our methods of data collection and our research aims clear.

18. Please describe when and how you reached saturation? How did you detect this?

We have added the text below to add further detail:

Methods:

Data saturation was deemed to have been achieved through thematic and code-saturation (18), which was discussed and determined between KS and LM.

Results:

Of 782 questionnaire respondents who provided consent to be contacted about an interview, we interviewed 23 (18 of whom were in paid work immediately prior to the start of the first UK lockdown and therefore included in this qualitative analysis) and reached data saturation at this stage.

19. looks like a convergent mixed method analysis? I miss information regarding the clear method.

Just to emphasise this is a mixed method study and when we revised the manuscript, we have tried to ensure that the approach is clear. As described within the Methods section, the study is “*a qualitative study embedded within an observational questionnaire study of individuals with musculoskeletal (MSK) conditions...*” We collected quantitative and qualitative data concurrently. For the purposes of the presented analysis, we have used the quantitative data on work status to describe the context of changes to working made during the pandemic. The collection of quantitative data provided this contextual information and facilitated sampling for the qualitative study. Please also see response to points 4 and 9 above.

20. written? or oral?

We have added the following detail:

Within “questionnaire data collection”: Written consent was obtained at the time of survey completion.

Within “qualitative data collection”: Consent was obtained prior to interview using a written consent guide and this was audio recorded. Interviews were audio recorded and transcribed verbatim.

21. See comments in table 2:

Table 2 Comments:

22. These themes are not described as 'meaningful' themes, (Table 2)

On reflection we appreciate why there may have been some confusion around the nomenclature used. The themes we have described are concordant with our analysis, which, as described in the methods was informed by a deductive and inductive approach. To aid clarity, we have renamed the first column “primary areas of inquiry” to reflect that these were deductive and mapped to the topic guide and have renamed our “sub-themes” as “themes”.

23. Not all the themes stand on their own. Some could be reformulated. (Table 2)

We hope that the changes we have made in response to point 24 clarify how we have conceptualised our areas of inquiry and themes. The themes we describe reflect aspects which informed individual changes to work status/circumstances and the overarching consequences of these changes; as well as the specific advantages and disadvantages of changes that were made to working during the pandemic.

24. It is unusual that there is a mix of the data from the questionnaire and the interviews.

The free-text responses to open ended questions collected within the questionnaire often described changes that had happened to individuals’ working circumstances and therefore directly addressed the aims of the study. Such responses were often lengthy and detailed and provided a large amount of rich and insightful data that could be analysed thematically in addition to the interview data. Free text comments are often provided by respondents but then ignored and it is unethical to ask people to give responses which are not then used (O’Cathain & Thomas, 2004). Whilst analysis of free text responses is not a substitute for well conducted qualitative research, it is another form of qualitative data that can be analysed like any other form of qualitative data, offering further illustrations and confirmation of findings from interviews (Garcia et al., 2012).

Other studies have similarly combined open ended questions responses/online forum responses with interview data for thematic analysis e.g., Sloan et al 2020 (COVID-19 and shielding: experiences of

UK patients with lupus and related diseases (nih.gov) conducted a thematic analysis of online forum posts and interview data.

O’Cathain & Thomas, 2004: <https://doi.org/10.1186/1471-2288-4-25>

Garcia et al., 2012: <https://link.springer.com/content/pdf/10.1023/B:QUQU.0000019394.78970.df>

25. Please be careful with generalisations while discussing qualitative research. You could rephrase this in another way.

Thank you, we have rephrased as follows:

People who continued working often reported significant anxiety about becoming infected at work and perceived unfairness compared to those who were furloughed, or could work at home.

26. concrete?

If this is referring to providing specific figures, these have been added to the discussion.

27. 2 totally different situations during and after lockdown. Please indicate the possible bias in more details.

We have provided further detail about the timing of data collection.

28. There is no methodological discussion added to this chapter. Please add a chapter, where you discuss the + and - of the approach.

We have discussed the strengths and weaknesses of the study methods within the second paragraph of the discussion (starting ‘However, there are some limitations to consider when interpreting these findings.’), and these methodological limitations include collecting data at a single point in time from each participant and having a high number of participants who were in professional/associate professional occupations. We have rephrased the start of the second paragraph in the discussion to make it clear it is discussing the strengths and weaknesses of the study. We have structured the discussion as per the BMJ proposed structured discussion: summary of key findings, methodological strengths and limitations, comparison with existing literature, conclusions including implications for policy and practice. Please also see response to point 32 below.

29. language - we is not scientific.

The use of the first person, including “we”, is not uncommon in academic writing when referring to your own body of work (*cf* Sloan et al., Rheumatology 2022; Smith et al., BMJ Open 2021; Hollick et al., ARD 2020);

<https://doi.org/10.1093/rheumatology/keab937>

<http://dx.doi.org/10.1136/bmjopen-2021-048772>

<http://dx.doi.org/10.1136/annrheumdis-2020-216988>

In addition, the passive voice does not represent the way qualitative research is an active process. We have checked the style of other BMJ publications, and this appears to be consistent with the style used.

30. In general: I miss some structure in the discussion. Please group the subject. start with methodological issues, following by the discussion of the findings, etc.

We have structured the discussion as per the BMJ proposed structured discussion; summary of key findings, methodological strengths and limitations, comparison with existing literature, conclusions including implications for policy and practice. Following BMJ Open Guidance below, subheadings are not a requirement of the Discussion section and we believe our Discussion section is concordant with the sections proposed in this guidance:

“We also recommend, but do not insist, that the discussion section is no longer than five paragraphs and follows this overall structure (you do not need to use these as subheadings): a statement of the principal findings; strengths and weaknesses of the study; strengths and weaknesses in relation to other studies, discussing important differences in results; the meaning of the study: possible explanations and implications for clinicians and policymakers; and unanswered questions and future research.” (Authors - BMJ Open)

Reviewer 2 (Dr. Fatih Karaarslan, University of Health Sciences, Ankara):

General remarks:

31. It is not understandable and some abbreviations need to be explain (e.g. MSK in abstract). It could be better to rearrange whole manuscript. Kind regards

We have spelled out MSK in the abstract. In making the revisions to the manuscript we have ensured that the structure is consistent with BMJ open recommendations, and the discussion is in the proposed structured format of the BMJ. We have made changes to improve the understanding of methods and clarity of results as per the Editor’s and Reviewer 1’s comments and we hope these changes address your concerns.

VERSION 2 – REVIEW

REVIEWER	Stijn De Baets Ghent University, Department of rehabilitation sciences
REVIEW RETURNED	23-Feb-2022

GENERAL COMMENTS	Thank you for the answers to questions and the revision of the manuscript.
--